# VisualQuality-R1: Reasoning-Induced Image Quality Assessment via Reinforcement Learning to Rank

**Tianhe Wu**[1,2], **Jian Zou**[1], **Jie Liang**[2], **Lei Zhang**[2,3],*and **Kede Ma**[1]*

[1]City University of Hong Kong
[2]OPPO Research Institute
[3]The Hong Kong Polytechnic University
{tianhewu-c, jian.zou}@my.cityu.edu.hk, liang27jie@gmail.com,
cslzhang@comp.polyu.edu.hk, kede.ma@cityu.edu.hk
https://github.com/TianheWu/VisualQuality-R1

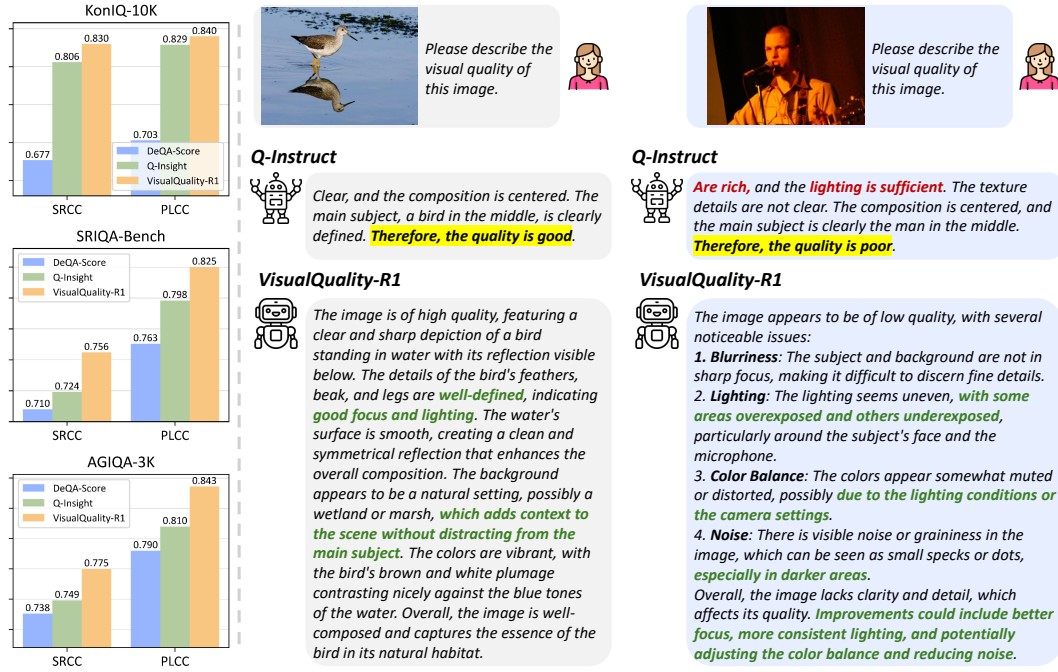

Figure 1: VisualQuality-R1 excels at image quality scoring, while generating contextually rich, human-aligned quality descriptions.

## Abstract

DeepSeek-R1 has demonstrated remarkable effectiveness in incentivizing reasoning and generalization capabilities of large language models (LLMs) through reinforcement learning. Nevertheless, the potential of reasoning-induced computation has not been thoroughly explored in the context of image quality assessment (IQA), a task depending critically on visual reasoning. In this paper, we introduce VisualQuality-R1, a reasoning-induced no-reference IQA (NR-IQA) model, and we train it with reinforcement learning to rank, a learning algorithm tailored to the intrinsically relative nature of visual quality. Specifically, for a pair of images, we employ group relative policy optimization to generate multiple quality scores for each image. These estimates are used to compute comparative probabilities

---

*Corresponding authors.

39th Conference on Neural Information Processing Systems (NeurIPS 2025).

of one image having higher quality than the other under the Thurstone model. Rewards for each quality estimate are defined using continuous fidelity measures rather than discretized binary labels. Extensive experiments show that the proposed VisualQuality-R1 consistently outperforms discriminative deep learning-based NR-IQA models as well as a recent reasoning-induced quality regression method. Moreover, VisualQuality-R1 is capable of generating contextually rich, human-aligned quality descriptions, and supports multi-dataset training without requiring perceptual scale realignment. These features make VisualQuality-R1 especially well-suited for reliably measuring progress in a wide range of image processing tasks like super-resolution and image generation.

# 1 Introduction

Image quality assessment (IQA) aims to quantify the visual quality of digital images consistent with human perceptual judgments. Commonly, IQA models are classified into full-reference (FR) and no-reference (NR) approaches [47], depending on the availability of pristine-quality reference images. In this paper, we focus on NR-IQA due to its practical relevance in real-world scenarios where reference images are unavailable. Over the decades, NR-IQA has evolved from knowledge-driven [33, 12] to data-driven approaches [30, 19, 54], and shifted from regression-based to ranking-based [58, 59] techniques. Nevertheless, achieving strong model generalization (*e.g.*, generalization to unseen image distortions) remains a significant, unresolved challenge, driving recent research toward multi-dataset training [6], active fine-tuning [44], and continual model adaptation [57].

The rapid advancement of vision-language models (VLMs) offers promising avenues for enhancing NR-IQA generalization by contextualizing it into broader vision tasks [51]. VLMs can effectively integrate multi-modal information, enabling understanding of both low-level image distortions (*e.g.*, noise and blur) and high-level perceptual attributes (*e.g.*, aesthetics and content semantics). This multi-modal semantic contextualization allows VLMs to articulate nuanced quality descriptions with stronger generalization.

However, current NR-IQA methods mainly leverage VLMs through supervised fine-tuning (SFT), which face several critical limitations [49, 56]. First, constructing informative quality descriptions demands extensive human effort, rendering the annotation process labor-intensive and expensive[2]. Second, models trained via SFT often overfit to the biases and idiosyncrasies present in training data, and may unintentionally encounter catastrophic forgetting of acquired knowledge during pre-training. Third, SFT typically yields overly rigid and templated outputs (see Fig. 1) that may be less useful.

Reinforcement learning (RL) has recently emerged as a powerful alternative, enhancing the reasoning capabilities of LLMs, while aligning their responses with human preferences [35, 13]. In particular, DeepSeek-R1 [13] demonstrates the effectiveness of RL in promoting generalization by encouraging automated exploration of plausible reasoning paths and employing rule-based rewards to prevent reward hacking [37]. However, a direct adaptation of RL techniques to NR-IQA, as exemplified by the recent Q-Insight model [21], has been limited by its reliance on dataset-specific reward design and additional distortion-type classification. These constraints stem from its treatment of visual quality as an *absolute* perceptual quantity, thereby framing NR-IQA naïvely as a regression task.

In this paper, we introduce **VisualQuality-R1**, a reasoning-induced NR-IQA model, and we train it via reinforcement learning to rank (RL2R), a learning algorithm explicitly designed to capture the inherently *relative* nature of visual quality. Specifically, we employ group relative policy optimization (GRPO) [36] to derive multiple quality scores for each image in a pair. We then compute comparative probabilities between images using the Thurstone model [41] by assessing the difference between the mean quality score of one image and individual quality scores of another, normalized by their sample variances. Unlike previous methods, we define the reward function using the continuous fidelity measure [42], which provides precise guidance to facilitate quality ranking. Extensive experiments confirm that VisualQuality-R1 effectively assesses visual quality across a diverse range of distortion scenarios, outperforming discriminative deep learning-based NR-IQA models as well as a recent reasoning-induced quality regression method [21]. Moreover, VisualQuality-R1 generates

---

[2]Utilizing state-of-the-art proprietary VLMs such as GPT-4o [16] for automated annotation suffers from similar scalability challenges due to high computational costs and financial burdens.

contextually rich, human-aligned quality descriptions (see Fig. 1), which can be leveraged to provide targeted feedback for downstream image processing algorithms, and support fine-grained quality control in digital photography pipelines. Additionally, we demonstrate that VisualQuality-R1 remains effective across multi-dataset training scenarios without requiring perceptual scale realignment.

## 2 Related Work

This section provides a structured review of related NR-IQA models, emphasizing recent advancements, particularly those leveraging VLMs.

**Regression-based Models** NR-IQA models primarily employed regression-based approaches, wherein image quality was treated as an absolute perceptual quantity directly estimated from extracted "quality-aware" features. Initially, features were handcrafted based on natural scene statistics [33, 34], degradation-specific characteristics [46, 48, 24], and perceptual models inspired by the human visual system [45]. Nonetheless, these methods were limited by the representational capacity of handcrafted features. Later, deep learning-based regression models emerged as the dominant paradigm, using end-to-end trainable neural networks to directly predict quality scores (or, in some cases, quality distributions) [18, 30, 3, 39, 54, 52]. These models typically utilize standard regression losses such as the mean squared error and mean absolute error, or statistical distances such as the earth mover's distance [39] and Kullback–Leibler (KL) divergence [55]. Regression-based models often struggle with generalization issues, and require labor-intensive perceptual scale realignment [31] when training on multiple IQA datasets.

**Ranking-based Models** To address these shortcomings, ranking-based NR-IQA models were introduced, modeling visual quality as an intrinsically relative perceptual quantity. Gao *et al.* [10] pioneered the concept of quality ranking in NR-IQA, although their initial implementation relied on predefined anchor images and was not end-to-end optimized. Ma *et al.* [29] adapted RankNet [4] to NR-IQA by training (though not fully end-to-end) on quality-discriminable image pairs. Their subsequent work established the first end-to-end ranking-based NR-IQA method grounded in the Thurstone model [41]. Nevertheless, their approach suffers from scaling ambiguity during variance estimation. Zhang *et al.* [58] incorporated a hinge loss to regularize variance estimation, yet the scaling ambiguity persisted. Their study also demonstrated the superiority of the fidelity loss [42] over the conventional cross-entropy loss in ranking-based NR-IQA. Subsequent research has adopted a simpler approach by fixing the variance parameter to one (corresponding to the Thurstone Case V model), facilitating active fine-tuning of NR-IQA models on challenging examples [44, 43] and allowing for continual adaptation to novel distortion scenarios [57]. Other losses that enable quality ranking include the margin ranking loss [25], differentiable approximations of Spearman's rank correlation coefficient (SRCC) [2], Pearson linear correlation coefficient (PLCC) [53], and statistical distances between permutation probabilities [5, 38, 17].

**VLM-based Models** The integration of VLMs into NR-IQA has recently gained traction, particularly due to their proficiency in capturing contextual semantics through multi-modal representation learning. Early attempts include multitask adaptation of CLIP [59], as well as SFT-based methods like Q-Align [50], Compare2Score [60], DepictQA [56], and DeQA-Score [55], which trained VLMs to generate either quality scores, distributions, or descriptions. Closest to ours, Q-Insight [21] explored reasoning-induced quality regression through RL. However, Q-Insight struggles with the dataset-specific reward calibration, added complexity of auxiliary distortion-type classification, and generalization to novel distortion scenarios. In contrast, our VisualQuality-R1 redefines the use of VLMs in NR-IQA by shifting from absolute regression to relative ranking, leading to enhanced generalization across distortion scenarios with better quality justifications.

## 3 Reasoning-Induced NR-IQA

To harness both the powerful reasoning-inducing capabilities of RL and the intrinsically relative nature of visual quality, we propose an NR-IQA model—VisualQuality-R1—and an RL2R method of training it that seamlessly integrates the Thurstone model within GRPO. Fig. 2 shows the system diagram of VisualQuality-R1.

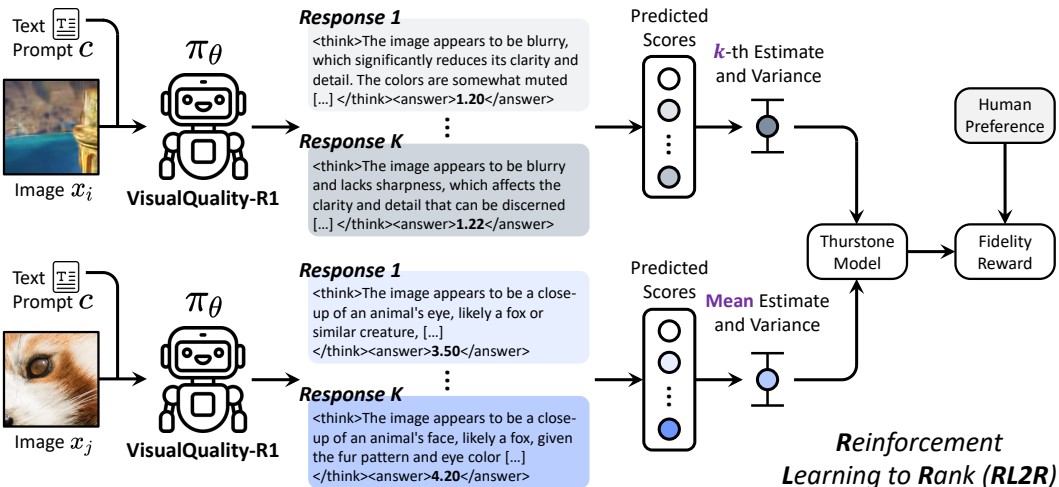

Figure 2: System diagram of the proposed VisualQuality-R1 trained via RL2R. Given an image pair $(x_i, x_j)$ with a shared text prompt $c$, VisualQuality-R1 generates $K$ responses. Following GRPO [36], each response includes a detailed reasoning process and a predicted quality score. To assess relative visual quality, we calculate the asymmetric comparative probability that image $x_i$ is perceived better than $x_j$ under the Thurstone model [41]. This involves subtracting the mean predicted score of $x_j$ from the $k$-th score of $x_i$, standardized by their sample variances. A fidelity reward is derived from human preference, providing continuous supervisory signals for policy optimization.

### 3.1  VisualQuality-R1 via RL2R

Given a text prompt $c$ and an image $x$, our goal is to fine-tune a pre-trained VLM, with policy $\pi_\theta(\cdot|c, x)$, to produce a scalar quality score in the range of $[1, 5]$, following a step-by-step reasoning process, encapsulated within specially designated tags for explicit instruction and enhanced interpretability. The complete structured text prompt is provided in Table 1.

More specifically, for a training batch of images $\{x_1, x_2, \ldots, x_B\}$, where $B$ is the minibatch size, we apply GRPO to generate $K$ quality predictions for $x_i$, $q(x_i) = [q_1(x_i), q_2(x_i), \ldots, q_K(x_i)]^\mathsf{T}$. This output naturally encodes predictive uncertainty, which is crucial for making reliable relative quality ranking. Under the Thurstone model [41], the visual quality of an image is assumed to follow a Gaussian distribution. Thus, we compute the asymmetric comparative probability for each of the $B \times (B - 1)$ ordered image pairs by subtracting the mean quality score of $x_j$ from the $k$-th quality score of $x_i$, standardized by their sample variances:

$$p_k(x_i, x_j) = \Phi\left(\frac{q_k(x_i) - \mu(q(x_j))}{\sqrt{\sigma^2(q(x_i)) + \sigma^2(q(x_j)) + \gamma}}\right), \quad \text{for } i \neq j, \tag{1}$$

where $\Phi(\cdot)$ is the standard Gaussian cumulative distribution function. $\mu(q(x_j))$ and $\sigma^2(q(x_j))$ represent the mean and variance of the quality predictions for $x_j$, respectively. $\gamma$ is a small positive constant to avoid any potential division by zero. Compared to previous ranking-based NR-IQA models that fix the variance parameter in Eq. (1) to one, we explicitly leverage sample variances derived from GRPO. This gives us an opportunity to dynamically accommodate predictive uncertainty for different images. Meanwhile, using the sample mean for quality comparison stabilizes the asymmetric probability estimate and the subsequent reward calculation by appropriately penalizing outlier predictions.

The true preference $p(x_i, x_j)$ is derived from human mean opinion scores (MOSs):

$$p(x, y) = \begin{cases} 1 & \text{if } \mathrm{MOS}(x) > \mathrm{MOS}(y) \\ 0.5 & \text{if } \mathrm{MOS}(x) = \mathrm{MOS}(y) \\ 0 & \text{otherwise} \end{cases}. \tag{2}$$

An important aspect of our RL2R algorithm is that we define the reward function $r_k(x_i)$ for each quality estimate $q_k(x_i)$ as the fidelity measure [42]—a continuous analogue of the discretized binary

Table 1: Structured text prompt used in VisualQuality-R1.

---

You are doing the image quality assessment task. Here is the question:
What is your overall rating on the quality of this picture? The rating should be a float between 1 and 5, rounded to two decimal places, with 1 representing very poor quality and 5 representing excellent quality. First output the thinking process in `<think>` `</think>` tags and then output the final answer with only one score in `<answer>` `</answer>` tags.

---

reward [13, 21], averaged across all $B - 1$ image pairs:

$$r_k(x_i) = \frac{1}{B-1} \sum_{j \neq i} \left( \sqrt{p(x_i, x_j)p_k(x_i, x_j)} + \sqrt{(1 - p(x_i, x_j))(1 - p_k(x_i, x_j))} \right). \quad (3)$$

This continuous reward feedback provides precise guidance during RL2R by capturing subtle variations in quality ranking, thus improving generalization across diverse distortion scenarios. We collect $K$ fidelity rewards for $x_i$ into the vector $r(x_i) = [r_1(x_i), r_2(x_i), \ldots, r_K(x_i)]^\mathsf{T}$, and compute the relative advantage $a_k(x_i)$ by standardizing rewards within group:

$$a_k(x_i) = \frac{r_k(x_i) - \mu(r(x_i))}{\sigma(r(x_i))}. \quad (4)$$

The final policy update of $\pi_\theta(\cdot|c, x_i)$ is guided by the regularized objective in GRPO:

$$\ell(\theta) = \frac{1}{BK} \sum_{i=1}^{B} \sum_{k=1}^{K} \left( \min \left( \frac{\pi_\theta(o_k|c, x_i)}{\pi_{\theta_{old}}(o_k|c, x_i)} a_k(x_i), \mathrm{clip} \left( \frac{\pi_\theta(o_k|c, x_i)}{\pi_{\theta_{old}}(o_k|c, x_i)}, 1 - \epsilon, 1 + \epsilon \right) a_k(x_i) \right) \right.$$
$$\left. - \beta\, D_{\mathrm{KL}} \left( \pi_\theta(o_k|c, x_i) \| \pi_{\mathrm{ref}}(o_k|c, x_i) \right) \right). \quad (5)$$

Here, $\pi_{\theta_{\mathrm{ref}}}(\cdot|c, x_i)$ denotes the stable reference policy obtained after VLM pre-training, and $\pi_{\theta_{old}}(\cdot|c, x_i)$ is the policy from the previous RL2R training epoch, from which we sample $K$ reasoning trajectories $o = \{o_k\}_{k=1}^{K}$. The second KL divergence term is approximated by

$$D_{\mathrm{KL}} \left( \pi_\theta(o_k|c, x_i) \| \pi_{\mathrm{ref}}(o_k|c, x_i) \right) = \frac{\pi_{\mathrm{ref}}(o_k|c, x_i)}{\pi_\theta(o_k|c, x_i)} - \log \frac{\pi_{\mathrm{ref}}(o_k|c, x_i)}{\pi_\theta(o_k|c, x_i)} - 1, \quad (6)$$

incorporated to ensure that the updated policy $\pi_\theta(\cdot|c, x_i)$ does not deviate excessively from $\pi_{\mathrm{ref}}(\cdot|c, x_i)$. $\epsilon$ is the clipping threshold to prevent large and potentially destabilizing updates to the policy. The coefficient $\beta$ serves as a balancing factor between the reward-weighted likelihood term and the KL regularization term.

We conclude this section by highlighting the key strengths of our VisualQuality-R1. First, VisualQuality-R1 inherits all the advantages of ranking-based NR-IQA models, enabling effective multi-dataset training, active fine-tuning, and continual model adaptation without requiring perceptual scale realignment [31], a feature notably absent in regression-based NR-IQA approaches. Second, trained via RL2R, VisualQuality-R1 mitigates the scalability and overfitting issues inherent in SFT-based models. Third, VisualQuality-R1 promises to both improve model generalizability and furnish contextually rich textual justifications alongside numerical quality scores, thereby boosting its practical relevance in real-world IQA applications.

## 4 Experiments

To validate VisualQuality-R1, we conduct comprehensive experiments across diverse distortion scenarios, ablation studies on key design components, and in-depth analysis of model behaviors.

### 4.1 Experimental Setups

**Competing Models and Training Details** Competing methods encompass three categories: 1) handcrafted models: NIQE [34] and BRISQUE [33]; 2) discriminative deep-learning-based models:

Table 2: SRCC and PLCC results of NR-IQA models trained on KADID-10K. Exceptions include Q-Insight[†] and VisualQuality-R1[†], which use a combined training set (KADID-10K and SPAQ). Top two results are highlighted in **bold** and underline, respectively.

| Method | Imaging-Related Distortion | | | | Processing-Related Distortion | | | | Avg |
|---|---|---|---|---|---|---|---|---|---|
| | BID | CLIVE | KonIQ | SPAQ | De-blurring | Super-Res. | De-hazing | Image Gen. | |
| **SRCC** | | | | | | | | | |
| *Handcrafted* | | | | | | | | | |
| NIQE [34] | 0.515 | 0.450 | 0.421 | 0.676 | 0.360 | 0.557 | 0.343 | 0.533 | 0.482 |
| BRISQUE [33] | 0.522 | 0.314 | 0.385 | 0.614 | 0.389 | 0.482 | 0.242 | 0.497 | 0.431 |
| *Discriminative Deep-Learning-based* | | | | | | | | | |
| UNIQUE [58] | 0.412 | 0.470 | 0.649 | 0.751 | 0.669 | 0.649 | 0.577 | 0.608 | 0.598 |
| MUSIQ [19] | 0.327 | 0.284 | 0.473 | 0.720 | 0.656 | 0.404 | 0.458 | 0.494 | 0.477 |
| MANIQA [54] | 0.420 | 0.487 | 0.213 | 0.745 | 0.726 | 0.263 | 0.608 | 0.422 | 0.486 |
| *VLM-based* | | | | | | | | | |
| LIQE [59] | 0.677 | 0.719 | 0.684 | 0.815 | 0.797 | 0.743 | **0.646** | 0.653 | 0.717 |
| Q-Align [50] | 0.576 | 0.554 | 0.573 | 0.767 | 0.761 | 0.684 | 0.455 | 0.682 | 0.632 |
| DeQA-Score [55] | 0.702 | 0.743 | 0.677 | 0.852 | 0.785 | 0.710 | 0.643 | 0.738 | 0.731 |
| Qwen2.5-VL-7B [1] | 0.711 | 0.733 | 0.754 | 0.848 | 0.820 | 0.603 | 0.458 | 0.735 | 0.708 |
| Q-Insight [21] | 0.784 | 0.761 | 0.806 | 0.872 | 0.831 | 0.724 | 0.601 | 0.749 | 0.766 |
| Q-Insight[†] | 0.806 | 0.804 | 0.812 | 0.907 | **0.846** | 0.700 | 0.539 | 0.657 | 0.759 |
| VisualQuality-R1 | 0.790 | 0.750 | 0.830 | 0.875 | 0.838 | **0.756** | 0.598 | **0.775** | 0.777 |
| VisualQuality-R1[†] | **0.811** | **0.811** | **0.855** | **0.913** | 0.845 | 0.752 | 0.588 | 0.754 | **0.791** |
| **PLCC** | | | | | | | | | |
| *Handcrafted* | | | | | | | | | |
| NIQE [34] | 0.527 | 0.494 | 0.439 | 0.683 | 0.376 | 0.587 | 0.482 | 0.560 | 0.519 |
| BRISQUE [33] | 0.528 | 0.362 | 0.400 | 0.624 | 0.380 | 0.556 | 0.468 | 0.541 | 0.482 |
| *Discriminative Deep-Learning-based* | | | | | | | | | |
| UNIQUE [58] | 0.385 | 0.472 | 0.590 | 0.708 | 0.654 | 0.668 | 0.578 | 0.581 | 0.580 |
| MUSIQ [19] | 0.280 | 0.325 | 0.435 | 0.666 | 0.563 | 0.441 | 0.455 | 0.434 | 0.450 |
| MANIQA [54] | 0.512 | 0.571 | 0.257 | 0.753 | 0.728 | 0.243 | 0.663 | 0.448 | 0.522 |
| *VLM-based* | | | | | | | | | |
| LIQE [59] | 0.680 | 0.726 | 0.652 | 0.814 | 0.712 | 0.775 | 0.661 | 0.653 | 0.709 |
| Q-Align [50] | 0.651 | 0.643 | 0.612 | 0.779 | 0.802 | 0.713 | 0.525 | 0.705 | 0.679 |
| DeQA-Score [55] | 0.743 | 0.795 | 0.703 | 0.858 | 0.838 | 0.763 | **0.688** | 0.790 | 0.772 |
| Qwen2.5-VL-7B [1] | 0.725 | 0.760 | 0.810 | 0.854 | 0.852 | 0.653 | 0.553 | 0.810 | 0.752 |
| Q-Insight [21] | 0.796 | 0.795 | 0.829 | 0.872 | 0.857 | 0.798 | 0.669 | 0.810 | 0.803 |
| Q-Insight[†] | 0.818 | 0.837 | 0.809 | 0.912 | 0.861 | 0.779 | 0.626 | 0.705 | 0.793 |
| VisualQuality-R1 | 0.806 | 0.794 | 0.840 | 0.878 | 0.872 | **0.825** | 0.651 | **0.843** | 0.814 |
| VisualQuality-R1[†] | **0.820** | **0.844** | **0.870** | **0.917** | **0.879** | 0.824 | 0.674 | 0.820 | **0.831** |

UNIQUE [58], MUSIQ [19], and MANIQA [54]; 3) VLM-based models: LIQE [59], Q-Align [50], DeQA-Score [55], Q-Insight [21], as well as the pre-trained Qwen2.5-VL-7B [1] baseline.

We fine-tune Qwen2.5-VL-7B [1] as the backbone for VisualQuality-R1 using GRPO [36]. The AdamW optimizer [27] is employed with an initial learning rate of $1 \times 10^{-6}$ and a linear decay schedule. For GRPO, we generate six candidate responses per prompt (*i.e.*, $K = 6$) and set the balance coefficient $\beta$ to $0.04$. Training runs on 16 NVIDIA A100 GPUs with a minibatch size of eight per GPU, taking approximately five hours for a total of 10 epochs.

## 4.2 Main Results

**Single-Dataset Training** We first train NR-IQA models on the synthetic KADID-10K [23] training set ($6 : 2 : 2$ split while ensuring content independence) and test in a zero-shot setting across eight datasets with distortions arising from digital imaging and (post-)processing stages: BID [7], CLIVE [11], KonIQ-10k [15], SPAQ [8], Liu13 (deblurring) [26], SRIQA-Bench (super-resolution) [6], Min19 (dehazing) [32], and AGIQA-3K (image generation) [20].

The SRCC and PLCC results presented in Table 2 reveal several key observations. First, all VLM-based models outperform traditional and discriminative deep-learning-based ones, with the base Qwen2.5-VL-7B achieving an SRCC of $0.708$ despite no IQA-specific training. This underscores

Table 3: PLCC results of VisualQuality-R1 with varying $K$ in GRPO. The default setting is highlighted in **bold**.

| #Generated Responses | Imaging-Related Distortion | | | | Processing-Related Distortion | | | | Avg |
|---|---|---|---|---|---|---|---|---|---|
| | BID | CLIVE | KonIQ | SPAQ | De-blurring | Super-Res. | De-hazing | Image Gen. | |
| $K=4$ | 0.805 | 0.795 | 0.839 | 0.875 | **0.875** | 0.815 | 0.643 | **0.844** | 0.811 |
| $K=5$ | **0.806** | **0.804** | **0.840** | **0.879** | 0.867 | **0.826** | 0.639 | 0.840 | 0.813 |
| **K = 6** | **0.806** | 0.794 | **0.840** | 0.878 | 0.872 | 0.825 | **0.651** | 0.843 | **0.814** |

Table 4: Comparison of different Thurstone model variants [41] in GRPO of VisualQuality-R1.

| Method | Imaging-Related Distortion | | | | Processing-Related Distortion | | | | Avg |
|---|---|---|---|---|---|---|---|---|---|
| | BID | CLIVE | KonIQ | SPAQ | De-blurring | Super-Res. | De-hazing | Image Gen. | |
| SRCC | | | | | | | | | |
| Q-Insight [21] | 0.784 | **0.761** | 0.806 | 0.872 | 0.831 | 0.724 | 0.601 | 0.749 | 0.766 |
| Binary Reward | 0.780 | 0.756 | 0.821 | **0.877** | 0.834 | 0.748 | 0.587 | 0.771 | 0.772 |
| Probability Average (Eq. (7)) | 0.785 | **0.761** | **0.836** | 0.875 | 0.835 | 0.747 | 0.574 | **0.775** | 0.774 |
| Fixed Variance of One | 0.778 | 0.750 | 0.818 | 0.871 | 0.830 | 0.744 | **0.606** | 0.760 | 0.770 |
| VisualQuality-R1 | **0.790** | 0.750 | 0.830 | 0.875 | **0.838** | **0.756** | 0.598 | **0.775** | **0.777** |
| PLCC | | | | | | | | | |
| Q-Insight [21] | 0.796 | 0.795 | 0.829 | 0.872 | 0.857 | 0.798 | **0.669** | 0.810 | 0.803 |
| Binary Reward | 0.790 | 0.792 | 0.833 | 0.876 | 0.867 | **0.825** | 0.646 | 0.840 | 0.809 |
| Probability Average (Eq. (7)) | 0.796 | **0.797** | **0.844** | 0.875 | 0.861 | 0.817 | 0.621 | 0.831 | 0.805 |
| Fixed Variance of One | 0.791 | 0.785 | 0.817 | 0.873 | 0.852 | 0.802 | 0.655 | 0.818 | 0.799 |
| VisualQuality-R1 | **0.806** | 0.794 | 0.840 | **0.878** | **0.872** | 0.825 | 0.651 | **0.843** | **0.814** |

the power of current VLMs in capturing generalizable quality cues. Second, reasoning-induced models such as Q-Insight and VisualQuality-R1 surpass SFT-based counterparts like Q-Align and DeQA-Score. Third, the proposed VisualQuality-R1 achieves the best results on average, validating that RL2R aligns better with human perception of image quality than regression-based approaches.

**Multi-Dataset Training**   Our RL2R approach enables multi-dataset training without the need for perceptual scale realignment. To exploit this, we train VisualQuality-R1[†] on a combination of KADID-10K [23] and SPAQ [8] (again $6:2:2$ split while ensuring content independence). As shown in Table 2, VisualQuality-R1[†] yields consistent performance gains. Despite a minor dip in the image generation scenario, the average SRCC/PLCC rises from $0.777/0.814$ to $0.791/0.831$. In stark contrast, Q-Insight[†] [21] fails to benefit from multi-dataset training due to its inability to address perceptual scale variations[3]: KADID-10K uses ratings from 1 to 5, while SPAQ spans 0 to 100.

## 4.3   Ablation Studies

**Effect of $K$ in GRPO**   We vary the number of generated responses, $K$, while keeping all other settings fixed during GRPO. Table 3 shows that reducing $K$ from six (default) to four or five has only a marginal effect, offering a favorable trade-off between computational cost and accuracy.

**Binary Reward vs. Continuous Fidelity Reward**   Table 4 shows that, within the same RL2R framework, our continuous fidelity reward generalizes better than the binary reward adopted in GRPO [36]. Moreover, both reward variants consistently outperform the regression-based Q-Insight [21], underscoring the effectiveness of our RL2R optimization.

**Thurstone Model Variants**   To evaluate the effectiveness of mean quality computation in Eq. (1), we compare it with an alternative that averages probabilities across individual quality comparisons:

$$p_k(x_i, x_j) = \frac{1}{K} \sum_{k'=1}^{K} p_{k,k'}(x_i, x_j) = \frac{1}{K} \sum_{k'=1}^{K} \Phi\left( \frac{q_k(x_i) - q_{k'}(x_i)}{\sqrt{\sigma^2(q(x_i)) + \sigma^2(q(x_j)) + \gamma}} \right). \quad (7)$$

---

[3]To enable multi-dataset training of Q-Insight, we linearly rescale MOSs from different IQA datasets to $[1, 5]$, and apply a dataset-agnostic threshold to compute binary rewards.

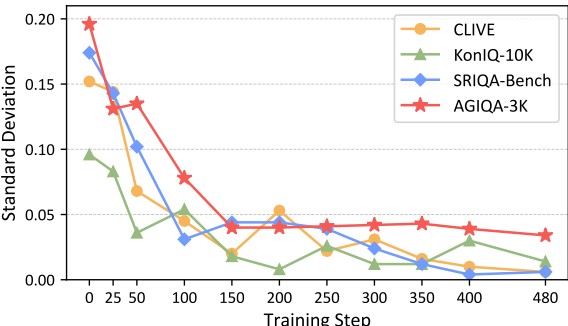

Figure 3: Prediction variability decreases during GRPO. We randomly select 20 images from each of CLIVE [11], KonIQ-10k [15], SRIQA-Bench [6], and AGIQA-3K [20]. At successive training steps, we generate multiple responses per image, compute the std of the predicted quality scores, and plot the average std across images. The uniformly downward trend confirms that VisualQuality-R1 becomes steadily more stable in assessing image quality as training progresses.

As reported in Table 4, averaging quality scores rather than probabilities yields higher performance across distortion scenarios, indicating more reliable comparative probability estimates and reward assignments. Taking a step further, we fix the variances in Eq. (7) to one—reducing the model to Thurstone Case V [41]. The constant-variance simplification degrades performance on nearly all datasets. This provides a strong indication that sample variances are capable of capturing the perceptual difficulty of image pairs, thus improving comparison reliability and stabilizing fidelity reward computation. Together, these findings verify that RL2R effectively embeds the Thurstone model within GRPO.

### 4.4 Further Analysis

**Predicted Score Variability over Iterations** We randomly sample 20 images from each of CLIVE [11], KonIQ-10k [15], SRIQA-Bench (super-resolution) [6], and AGIQA-3K (image generation) [20], respectively. At successive training checkpoints, we generate multiple responses per image and compute the standard deviation (std) of the resulting $K$ quality scores. As illustrated in Fig. 3, the std falls steadily across all datasets, indicating that predictions of VisualQuality-R1 become progressively more stable and confident.

**Visual Reasoning Evolution over Iterations** Fig. 4 tracks how the visual reasoning capabilities of VisualQuality-R1 mature over the course of training. The test image is super-resolved by SwinIR [22], which contains subtle, processing-related artifacts, making it an informative probe. Q-Insight notices that the image is "blurry" and "overexposed," but assigns an extremely low score (*i.e.*, 2.00), indicating limited sensitivity to super-resolution artifacts. The base model Qwen2.5-VL-7B [1] swings to the opposite extreme: it praises the "clear details" and "vibrant colors," declares the absence of blur or noise, and outputs an inflated score (*i.e.*, 4.80). The model clearly over-trusts superficial sharpness cues and misses hidden processing traces. In contrast, the proposed VisualQuality-R1 progressively refines its visual reasoning over iterations. At the 50-th step, it starts to suspect artificial stylization and questions the image's realism, yet it still values the apparent clarity. By the 200-th step, the description becomes more balanced. It acknowledges the level of detail and clarity, yielding a slightly higher but still cautious rating. At the last step, the explanation is now decidedly nuanced. VisualQuality-R1 attributes the remaining softness to possible filtering or to the object's inherent structure, labels the appearance "surreal," and reduces the score to 3.00, reflecting a judicious penalty for unnatural post-processing. In summary, RL2R guides VisualQuality-R1 from naïve, superficial remarks to sophisticated, human-aligned reasoning that correctly identifies subtle super-resolution artifacts and calibrates quality scores accordingly.

## 5 Conclusion and Discussion

We have introduced VisualQuality-R1, a reasoning-induced NR-IQA model optimized via RL2R. Our approach is grounded in the intrinsic relativity of visual quality, seamlessly integrating the Thurstone model within GRPO to capture predictive uncertainty. By introducing the continuous fidelity reward, VisualQuality-R1 delivers more precise policy-gradient signals.

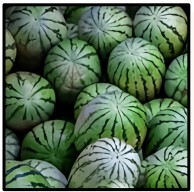

**Q-Insight**

\<think\>**The image appears to be somewhat blurry**, which affects the clarity and detail that can be discerned. The colors are vibrant, but the lack of sharpness reduces the overall quality. There's also a slight **overexposure** in some areas, which might be due to lighting conditions or camera settings.\</think\>
\<answer\>2.00\</answer\>

**Qwen2.5-VL-7B** (**0**-th Step)

\<think\>The image appears to be of high quality with clear details and vibrant colors. The texture and patterns on the objects are well-defined, suggesting good resolution and lighting. **There are no visible distortions or blurriness** that would typically indicate lower quality. The contrast and saturation levels seem appropriate, enhancing the visual appeal without overwhelming the viewer.\</think\>\<answer\>4.80\</answer\>

**VisualQuality-R1** (**50**-th Step)

\<think\>The image appears to be somewhat **abstract or stylized**, which might not be ideal for a realistic quality assessment. However, it is clear and the **details are well-defined**, suggesting good resolution. The colors are vibrant and the contrast is high, which can be visually appealing but may not reflect naturalistic quality. The image does not contain any discernible text or identifiable objects that could provide further context.\</think\>\<answer\>3.50\</answer\>

**VisualQuality-R1** (**200**-th Step)

\<think\>The image appears to have a high level of detail and clarity, with distinct patterns and colors that are well-defined. However, there is a **slight lack of sharpness around the edges**, which might be due to the texture or the way the image was captured. The lighting seems even, and the colors are vibrant, contributing positively to the overall quality.\</think\>\<answer\>3.80\</answer\>

**VisualQuality-R1** (**Last** Step)

\<think\>The image appears to be a close-up of **some kind of fruit or vegetable with a patterned surface**. The colors are vibrant, but the image seems to have a filter applied that gives it a somewhat **surreal appearance**. **The details are not very sharp, which might be due to the nature of the subject or the processing of the image**. However, the saturation and contrast are high, making the image visually striking.\</think\>\<answer\>3.00\</answer\>

Figure 4: Evolution of the reasoning capabilities of VisualQuality-R1 on an image super-resolved by SwinIR [22]. Initially, VisualQuality-R1 overlooks artifacts and overestimates quality; at later stages, it progressively detects stylization, blur, and filtering effects, yielding more accurate quality scores and human-aligned textual justifications. Zoom in for improved visibility.

Extensive experiments validate that VisualQuality-R1 consistently surpasses strong discriminative deep learning-based methods and a reasoning-induced baseline. Notably, it bridges the performance gap between synthetic and realistic distortions, demonstrating robustness to dataset inductive biases and noise. In addition to quantitative improvements, VisualQuality-R1 generates contextually rich, human-aligned quality descriptions, which not only enhance transparency and interpretability but also facilitate user trust and post-hoc diagnosis in downstream tasks, such as content filtering, local enhancement prioritization, and quality-aware image retrieval.

**Limitations and Future Directions** Despite the generalization capabilities demonstrated by VisualQuality-R1, several limitations and promising research directions merit further discussion. First, as a specific case of test-time scaling, VisualQuality-R1 is slow, expensive, and memory-hungry; it may also compound early errors into confidently wrong predictions. It is thus desirable to incorporate sample-adaptive reasoning, rationale compression or distillation, and self-consistency sampling to make VisualQuality-R1 faster, cheaper, and more robust. Second, VisualQuality-R1 relies on a single, fixed text prompt for all images, regardless of the underlying distortion scenario or application context. Incorporating application-aware prompt adaptation, for example via learned prompt-tuning or dynamic template selection, could tailor VisualQuality-R1's reasoning and scoring to specific image processing tasks, therefore improving its flexibility and accuracy. Third, VisualQuality-R1 is currently formulated as an NR-IQA model, focusing solely on distorted inputs without access to pristine-quality counterparts. It is interesting to extend VisualQuality-R1 to a reference-based setting, which allows a (possibly corrupted) reference image [51]—potentially differing in resolution, color gamut, dynamic range, or bit depth—to serve as a flexible anchor for content fidelity. Last, we foresee adapting the proposed RL2R learning algorithm to other perceptual assessment tasks, including image aesthetics assessment [40], human age estimation [14], and perceptual similarity ranking [9]. Collectively, these promising directions aspire to foster more intelligent, transparent, and adaptable perceptual systems.

## Acknowledgments

This work was supported in part by the Hong Kong ITC Innovation and Technology Fund (9440379 and 9440390) and the PolyU-OPPO Joint Innovative Research Center.

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

# Appendix

## A Generalization Probing via gMAD Competition

The group maximum differentiation (gMAD) competition [28] is a model comparison framework designed to evaluate the generalization capability of computational models, particularly in scenarios where exhaustive ground-truth labeling is impractical. In this framework, models take turns serving as the defender, while the remaining models act as attackers. The attackers aim to identify image pairs that the defender assigns similar quality scores to, but which they themselves rate very differently. These "adversarial" pairs are subsequently assessed by human observers, allowing for an efficient and targeted examination of model strengths and weaknesses.

Applying gMAD to compare KADID-10K-trained VisualQuality-R1 and Q-Insight on KonIQ-10k [15] reveals a clear performance distinction. VisualQuality-R1 consistently uncovers perceptual inconsistencies in Q-Insight's predictions. Conversely, when subjected to attacks, VisualQuality-R1 demonstrates strong robustness, aligning closely with human perception of image quality. This dual capability—high aggressiveness in exposing flaws and strong resistance to adversarial challenges—underscores VisualQuality-R1's superior generalization, affirming its reliability for real-world IQA.

## B Model Complexity Comparison

We present a quantitative comparison of model complexity across representative NR-IQA methods. As shown in Table 5, VisualQuality-R1 incurs higher inference costs than discriminative models such as MUSIQ [19] and MANIQA [54], as well as small-scale VLMs like LIQE [59]. This increased computational demand, however, reflects a deliberate design choice to support fine-grained quality reasoning and robust quality rating across diverse distortion scenarios.

## C Additional Visual Reasoning Examples

We provide additional visual reasoning examples to illustrate the perceptual alignment of VisualQuality-R1 under both synthetic and realistic distortion scenarios. It should be noted that VisualQuality-R1 occasionally assigns quality scores outside the prescribed range of $[1, 5]$ (*e.g.*,

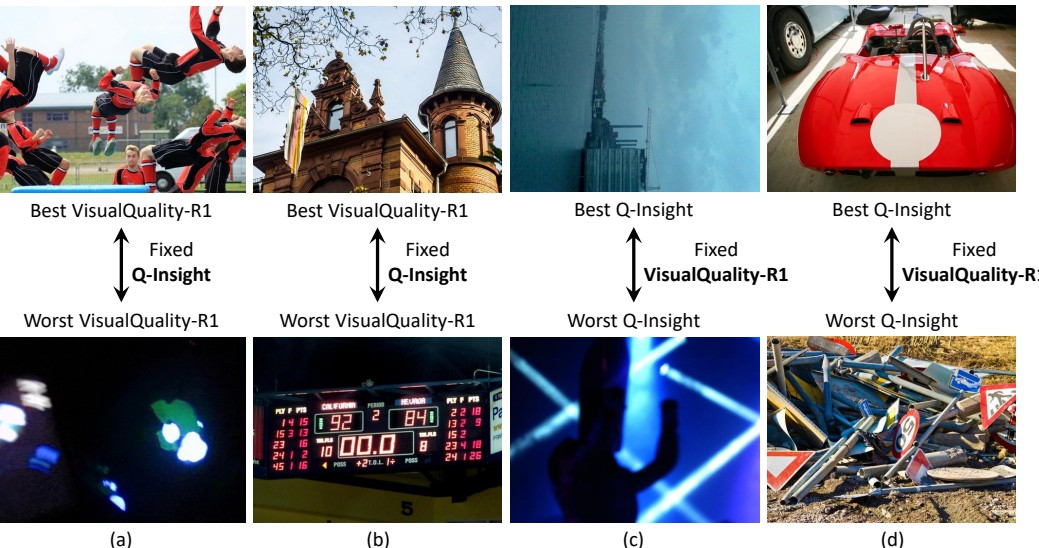

Figure 5: gMAD competition results between VisualQuality-R1 and Q-Insight [21]. (a) Fixed Q-Insight at the low-quality level. (b) Fixed Q-Insight at the high-quality level. (c) Fixed VisualQuality-R1 at the low-quality level. (d) Fixed VisualQuality-R1 at the high-quality level.

Table 5: Model complexity comparison using a $512 \times 384 \times 3$ image as input.

| Method | #Parameters | Inference Time | Inference Memory | #TFLOPs |
|---|---|---|---|---|
| UNIQUE [58] | 22.32 M | 0.02 s | 1.62 G | 0.029 |
| MUSIQ [19] | 27.13 M | 0.05 s | 1.69 G | 0.026 |
| MANIQA [54] | 135.75 M | 0.03 s | 2.13 G | 0.217 |
| LIQE [59] | 151.28 M | 0.03 s | 2.15 G | 0.131 |
| Q-Align [50] | 8.20 B | 0.14 s | 17.1 G | 1.98 |
| DeQA-Score [55] | 8.20 B | 0.11 s | 17.1 G | 1.98 |
| Q-Insight [21] | 8.29 B | 2.72 s | 17.6 G | 8.71 |
| VisualQuality-R1 | 8.29 B | 2.34 s | 17.6 G | 7.74 |

a score of $0.2$ for the second example in Fig. 6), which reflects a common limitation in LLMs. Nevertheless, these deviations have negligible impact on the overall quality prediction performance.

**Synthetic Distortion Scenario**

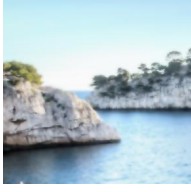

*<think>*
*The image appears to **be out of focus**, which significantly reduces its clarity and detail. **The edges of the rocks and trees are not sharp, and the water lacks definition**. This lack of focus affects the overall perception of the scene, making it difficult to discern finer details. The lighting seems adequate, but the blurriness overshadows any positive aspects.*
*</think>*
*<answer>1.00</answer>*

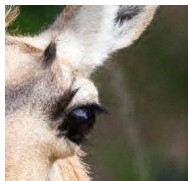

*<think>*
*The image appears to be a close-up shot of an animal's eye, likely a deer or similar species. **The focus is sharp on the eye, which is the main subject, but the background is blurred, indicating a shallow depth of field, which is often used to emphasize the subject**. The lighting seems natural, and there are no visible distortions or artifacts that would degrade the image quality. However, the image does not provide much context or additional details about the environment or the animal itself.*
*</think>*
*<answer>3.50</answer>*

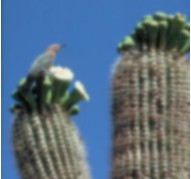

*<think>*
*The image appears to be **slightly out of focus**, which affects the sharpness and detail that can be seen. The colors are vibrant, but the lack of clarity reduces the overall quality. **The composition is interesting with the bird perched on the cactus, but the blurriness detracts from the visual impact**.*
*</think>*
*<answer>2.50</answer>*

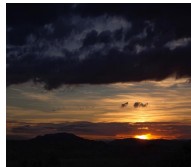

*<think>*
*The image captures a **beautiful sunset with a gradient of colors from dark blue at the top to warm oranges and yellows near the horizon**. The silhouette of the landscape adds depth and contrast. However, **the lower part of the image is quite dark, which might obscure some details**. The sky's texture and the subtle gradation of colors suggest good exposure for the upper part of the image. Overall, the image has a pleasing aesthetic but lacks detail in the darker areas.*
*</think>*
*<answer>3.80</answer>*

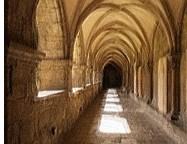

*<think>*
*The image appears to be somewhat **blurry and lacks sharpness**, which affects its clarity. The lighting seems adequate but not very bright, and there's a **noticeable graininess or noise** that reduces the overall quality. The composition is interesting with an architectural subject, but the technical aspects of the photo do not meet high standards.*
*</think>*
*<answer>2.80</answer>*

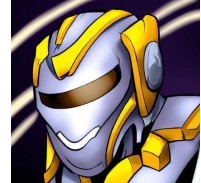

*<think>*
*The image appears to be **a digital illustration or a graphic design rather than a photograph**. It has **clear lines and vibrant colors**, which are typical for digital art. The details are sharp, and there's a good contrast between the elements, making it visually appealing. However, the lack of depth and realistic textures might suggest that it's not a high-resolution photograph but rather a stylized graphic. The overall impression is that it's a well-crafted image.*
*</think>*
*<answer>4.00</answer>*

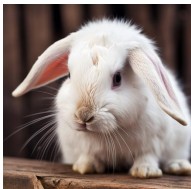

*<think>*
*The image is clear and well-lit, with good focus on the **rabbit's face and fur details**. The background is slightly blurred, which **helps to emphasize the subject**. The colors are natural, and there are no visible distortions or artifacts. Overall, the image quality appears to be high.*
*</think>*
*<answer>4.95</answer>*

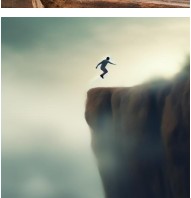

*<think>*
*The image appears to be **a digital artwork or a heavily edited photograph**. It features a silhouette of a person seemingly jumping off a cliff into a misty or foggy environment. The lighting is soft and diffused, which gives it a dreamlike quality. The focus is on the silhouette, making the background somewhat blurry, which is a common technique used in photography to emphasize the subject. **The colors are muted, and there's a lack of sharpness, which might be intentional for artistic effect.***
*</think>*
*<answer>2.80</answer>*

Figure 6: Visual reasoning samples from VisualQuality-R1 under the synthetic distortion scenario.

**Realistic Distortion Scenario**

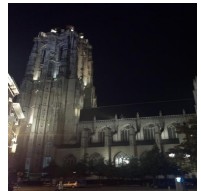

*<think>*
*The image appears to be **taken at night**, which can naturally reduce the overall quality due to lower light conditions. However, the building is well-lit, and the details of **the architecture are visible**. The exposure seems balanced, and there's no significant noise or blurriness that would degrade the image further. The composition is clear, and the subject (the building) is the focal point.*
*</think>*
*<answer>3.00</answer>*

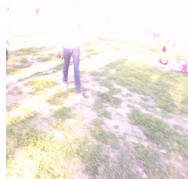

*<think>*
*The image appears to be **overexposed**, which means it is too bright and lacks detail. This makes it difficult to discern specific features or objects within the image. The lack of detail suggests that the image may not have been taken under ideal lighting conditions or that the **camera settings were not adjusted properly** for the scene.*
*</think>*
*<answer>0.20</answer>*

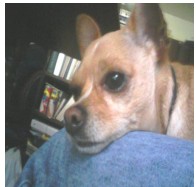

*<think>*
*The image appears to be somewhat **blurry**, which affects the sharpness and detail that could otherwise be present. The lighting seems adequate but not particularly bright or well-distributed, which might have contributed to some of the blurriness. The focus is not entirely clear, as **the edges of the dog's features are not sharply defined**. There is also a noticeable amount of noise or graininess in the image, which further reduces its quality.*
*</think>*
*<answer>1.80</answer>*

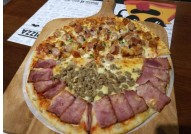

*<think>*
*The image is clear and well-lit, showing the **pizza in good detail**. The colors are vibrant, and the texture of the pizza toppings is visible. There are **no apparent distortions or blurs** that would affect the overall quality. The composition is centered, which helps focus attention on the subject.*
*</think>*
*<answer>4.87</answer>*

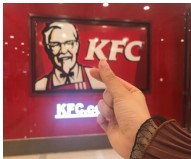

*<think>*
*The image appears to be clear and well-lit, with **good focus on the KFC logo and the hand in the foreground**. The colors are vibrant, and there is no noticeable noise or blurriness. However, the composition could be improved by centering the KFC logo more prominently in the frame. Despite this minor issue, the overall quality of the image is quite good.*
*</think>*
*<answer>4.20</answer>*

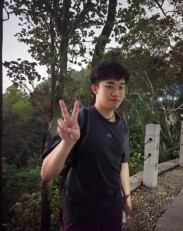

*<think>*
*The image appears to be clear and well-lit, **with good focus on the subject**. The background is slightly **out of focus but still recognizable as a natural setting with trees and a path**. The colors are natural and there are no visible distortions or artifacts. The composition is straightforward, with the subject centered and prominent.*
*</think>*
*<answer>4.38</answer>*

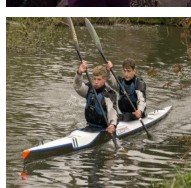

*<think>*
*The image is clear and well-lit, **capturing the action of two individuals kayaking**. The details of their **clothing and the kayak are visible, indicating good resolution**. The background is slightly out of focus, which helps to emphasize the subjects, but it does not detract significantly from the overall quality. There are no apparent distortions or blurs that would lower the quality.*
*</think>*
*<answer>4.70</answer>*

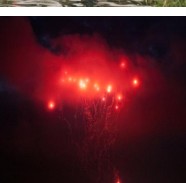

*<think>*
*The image appears to be taken at **night or in low-light conditions**, which is evident from the dark background and the presence of bright red lights. **The red lights seem to be part of fireworks or some other form of illumination, but the details are not clear due to the low light and the lack of focus**. There is also a significant amount of noise or graininess visible in the image, which further reduces its clarity. The composition is somewhat abstract, making it difficult to discern specific objects or events.*
*</think>*
*<answer>2.00</answer>*

Figure 7: Visual reasoning samples from VisualQuality-R1 under the realistic distortion scenario.

