# OpenReview forum: "VisualQuality-R1: Reasoning-Induced Image Quality Assessment via Reinforcement Learning to Rank"
_NeurIPS.cc/2025/Conference — NeurIPS 2025 spotlight_

### Official Review · Reviewer_5jDt · 2025-06-20

**Clarity:** 3
**Significance:** 3
**Originality:** 3
**Rating:** 4
**Confidence:** 3

**Summary:**

This paper proposes VisualQuality-R1, a no-reference IQA (NR-IQA) model that leverages reinforcement learning to rank (RL2R) with a novel integration of the Thurstone model and Group Relative Policy Optimization (GRPO). The core innovation lies in treating visual quality as a relative (rather than absolute) perceptual quantity.

**Questions:**

- The fixed prompt (Table 1) may hinder adaptation to context (e.g., synthetic art vs. natural images). Fig. C shows scores for stylized images lack nuance (e.g., rating digital art "4.00" without style sensitivity).
- GRPO’s \(K=6\) generations/image imply high costs. No latency/memory metrics are provided.
- Fig. B (supplement) shows an implausible score of **0.20** (below the 1–5 range)
- The conclusion suggests extending to FR-IQA but offers no experiments.

Finally, if the authors can actively address my concerns, I will be happy to improve the rating score.

**Ethical Concerns:**

["NO or VERY MINOR ethics concerns only"]

**Final Justification:**

Thank you for the author's response. I will maintain my original rating.

**Limitations:**

Adequately discussed (computational cost, prompt rigidity, error propagation in §9). The suggestion of "sample-adaptive reasoning" for efficiency is promising but untested.

**Paper Formatting Concerns:**

N.A.

**Quality:**

3

**Strengths And Weaknesses:**

As far as I  am concerened, **Strengths** are:  1) Strong empirical results: Comprehensive experiments (zero-shot generalization, multi-dataset training, ablation studies) validate superiority in SRCC/PLCC metrics (e.g., **0.791** avg. SRCC vs. 0.759 for Q-Insight).  2) Interpretability: Human-aligned textual explanations (Figs. B, C in supplement) enhance transparency, a significant advance over "black-box" NR-IQA models.  3)Practical impact: Enables multi-dataset training without MOS realignment, addressing a key bottleneck in real-world deployment.

**Weaknesses are:**

- Computational overhead: GRPO’s multiple generations per image (\(K=6\)) incur high inference costs. Efficiency trade-offs vs. gains are underexplored.
- Prompt rigidity: Reliance on a fixed text prompt (Table 1) limits adaptability to distortion-specific contexts (e.g., artistic vs. photographic images).
- Baseline depth: Comparisons with recent VLM-based methods (e.g., LLaVA-IQA) are missing. Q-Insight is the only RL-based baseline.
- Societal impact: While claimed as minimal (§9), potential misuse for automated content moderation (e.g., suppressing "low-quality" images) is unaddressed.

---

> ### Author Rebuttal · Authors · 2025-07-28
>
> We are deeply grateful for the reviewer’s thoughtful insights and thorough evaluation of our manuscript. Please find what below our detailed point-to-point responses to all the comments of this reviewer. We hope our responses can address this reviewer's concerns.
>
> ---
>
> **`[W1 & Q2] Regarding Computational Overhead`**
>
> Thanks for the constructive comment. We would like to clarify that during inference, we generate only one response per image (i.e., K=1), and the corresponding latency and memory usage metrics have been reported in our responses to **`Reviewer Cn1F [W4]`** and **`Reviewer NF6n [W1 & Q1]`**. Please refer to those sections for details.
>
> During training, we set the number of generations in GRPO to K=6. To better illustrate the **efficiency trade-offs versus performance gains**, we provide a detailed comparison of training cost and model performance under different generation numbers K=4,5,6. As shown in the table below, increasing K leads to a moderate and manageable increase in both GPU memory usage and training time. Meanwhile, the PLCC improves consistently from 0.811 at K=4 to 0.814 at K=6, suggesting that additional generations bring incremental yet stable gains.
>
> \begin{array}{lccc}
> \hline
>  & \text{K=4} & \text{K=5} & \text{K=6} \newline
> \hline
> \text{Memory} & 58\text{G per GPU} & 62\text{G per GPU} & 68\text{G per GPU} \newline
> \text{Time} & 3\text{h}39\text{min} & 4\text{h} 23\text{min} & 4\text{h} 45\text{min} \newline
> \text{Averaged PLCC} & 0.811 &0.813 & 0.814 \newline
> \hline
> \end{array}
>
> This empirical evidence further motivates our proposed direction of developing more efficient reasoning strategies, such as sample-adaptive generation or rationale distillation, as discussed in the section of **Limitations and Future Directions** in the main paper. We will include this clarification in the revised manuscript.
>
> ---
>
> **`[W2 & Q1] Regarding Prompt Rigidity`**
>
> Thanks for the valuable and constructive feedback. The limitation of using a fixed prompt has been honestly acknowledged in the section of **Limitations and Future Directions** (lines 252–256 in the main paper). To take a step forward, we designed a small set of scenario-specific prompts based on the original template in Table 1. For the detailed prompt settings, kindly refer to our responses to **`Reviewer NF6n [W2]`** . For the corresponding experimental results and analysis, please refer to our responses to **`Reviewer Cn1F [W5 & Q1]`**. In realistic scenarios, using **adaptive prompts at inference time leads to noticeable improvements**, highlighting the potential of our method to better adapt to distortion-specific contexts.
>
> Regarding the observation that *Fig. C shows scores for stylized images lack nuance (e.g., rating digital art "4.00" without style sensitivity)*, we thank the reviewer for pointing this out. We will include this as a representative failure case in the revised manuscript to illustrate the model’s limited sensitivity to artistic style when using fixed prompt. This also underscores the importance of designing scenario-specific prompts to better capture nuanced aspects such as artistic style, which we identify as a promising direction for future work.
>
> ---
>
> **`[W3] Regarding Baseline Depth`**
>
> Thanks for raising this important point. We agree that including comparisons with recent VLM-based methods is valuable. In fact, our main paper already includes several recent VLM-based NR-IQA baselines that are open-sourced and reproducible, such as LIQE, Q-Align, and DeQA-Score. These represent state-of-the-art approaches that leverage supervised fine-tuning on vision-language backbones.
>
> As for LLaVA-IQA, to the best of our knowledge, it is only briefly mentioned in a recent non-IQA-specific work ([1] VistaDream, ICCV 2025). To address this concern, we implemented a LLaVA-based baseline under the same evaluation protocol. As shown in the table below, LLaVA-IQA performs notably worse in terms of SRCC than VisualQuality-R1, particularly on distortion-focused datasets such as BID, CLIVE, and Deblurring. This underscores the challenge of adapting general-purpose VLMs to NR-IQA without task-specific training or reinforcement-based optimization. We will include this comparison in the revised manuscript.
>
> \begin{array}{l|cccc|cccc|c}
> \hline
> \text{Method} & \text{BID} & \text{CLIVE} & \text{KonIQ} & \text{SPAQ} &
> \text{Deblurring} & \text{Super-Resolution} & \text{Dehazing} &
> \text{Image Generation} & \text{Avg} \newline
> \hline
> \text{LLaVA-IQA}      & 0.446 & 0.401 & 0.437 & 0.672 & 0.620 & 0.633 & 0.488 & 0.612 &0.539    \newline
> \text{VisualQuality-R1} & 0.791  & 0.747  & 0.829  & 0.875 & 0.838  & 0.751  & 0.599 & 0.776  & 0.776 \newline
> \hline
> \end{array}
>
> [1] VistaDream: Sampling multiview consistent images for single-view scene reconstruction, ICCV 2025
>
> ---
>
> **`[W4] Regarding Societal Impact`**
>
> Thanks for raising this important point. While we initially assessed the societal impact as minimal (§9), we acknowledge that VisualQuality-R1 could be misused in automated content moderation to suppress images deemed “low-quality.” Although our model is designed to **support image enhancement and retrieval, not censorship**, we agree that such risks warrant explicit discussion.
>
> We will address this in the revised manuscript by highlighting the potential misuse and recommending safeguards, such as human-in-the-loop review and transparency about model limitations. We appreciate the reviewer for bringing attention to this important issue.
>
> ---
>
> **`[Q3] Regarding Implausible Score`**
>
> Thanks for pointing out this issue. The score of 0.20 in Fig. B indeed falls outside the instructed [1, 5] range, and we sincerely acknowledge this as an instruction-following failure, which is a common limitation observed in large-scale language models.
>
> In our experiments, we noticed that during the later training epochs (e.g., epochs 9 to 10), the model occasionally produced scores slightly below 1 or above 5. When training continued beyond epoch 10, these anomalies persisted, but they did not noticeably affect overall SRCC and PLCC performance, as both of which are relative ranking metrics.
>
> We believe there are two plausible reasons for this behavior:
> - Since the model is optimized with a learning-to-rank objective, it first learns to correctly rank images near the boundary scores (e.g., around 1 and 5). After this initial convergence, when it encounters samples that appear significantly worse than previous low-quality images (or significantly better than previous high-quality ones), it may exaggerate the relative differences by assigning scores below 1 or above 5 to maximize the fidelity-based reward.
> - More subtly, during policy optimization, the trade-off between reward maximization and KL regularization may not perfectly constrain the output head to obey soft output bounds. In rare cases, the model may exploit this slack to push scores slightly beyond the expected range.
>
> We will explicitly document this behavior in the revised manuscript and treat it as a known limitation. One potential remedy is to impose a soft or hard clipping mechanism to enforce outputs within the range without disrupting gradient flow.
>
> ---
>
> **`[Q4] Regarding The Extension to FR-IQA`**
>
> Thanks for pointing this out. We fully agree that extending VisualQuality-R1 to FR-IQA is a promising research direction, as also outlined in our Conclusion and Discussion section. In this rebuttal, we take a preliminary step toward this goal by adapting our model to FR-IQA settings and conducting initial experiments.
>
> We begin by modifying the prompt to include explicit reference comparison instructions. Specifically, we adopt the following prompt structure:
>
> *''... Here is the question: The first image serves as the reference for the second image. Please compare the second image against the reference and evaluate its overall visual quality. Provide a quality score as a float between 1.00 and 5.00, ...''*
>
> We fine-tuned VisualQuality-R1 on KADID-10K in the FR setting and evaluated it on three representative FR benchmarks: Deblurring, Super-Resolution, and Dehazing. The SRCC results are as follows. (Note: ''-FR'' uses both reference and distorted images; ''-NR'' uses only the distorted image.)
>
> \begin{array}{l|ccc}
> \hline
> \text{Method} &\text{Deblurring} & \text{Super-Resolution} & \text{Dehazing} \newline
> \hline
> \text{Qwen2.5-VL-7B-NR}      & 0.820 & 0.603 & 0.458 \newline
> \text{VisualQuality-R1-NR}      & 0.872 & 0.825 & 0.651 \newline \hline
> \text{Qwen2.5-VL-7B-FR}      & 0.751 & 0.488 & 0.389  \newline
> \text{VisualQuality-R1-FR}      & 0.869 & 0.767 & 0.581 \newline
> \hline
> \end{array}
>
> As shown in the above table, VisualQuality-R1-FR, fine-tuned with RL2R, significantly outperforms the zero-shot Qwen2.5-VL-7B in the full-reference setting. This demonstrates the effectiveness of our method in adapting VLM to the FR setting.
>
> However, we also note that VisualQuality-R1-FR still underperforms its NR counterpart. Similarly, in the zero-shot setting, Qwen2.5-VL-7B shows a notable drop in performance under FR. Upon further qualitative inspection, we found that in the FR setting, the model’s reasoning outputs frequently fail to capture subtle differences between the distorted and reference images. In some cases, the model even **incorrectly concludes that the two images are identical, despite the visible degradations**. These results suggest that reasoning over paired inputs in the FR setting is substantially more challenging and requires targeted learning strategies, which we leave for future work.
>
> These experiments serve as an early feasibility study toward extending VisualQuality-R1 to FR-IQA. We will include the results and analysis in the revised manuscript.

---

> ### Author Response · Authors · 2025-08-04
>
> Dear Reviewer 5jDt,
>
> Many thanks for your time in reviewing our paper and your constructive comments. We have submitted the point-to-point responses. We appreciate if you could let us know whether your concerns have been addressed, and we are happy to answer any further questions.
>
> Best regards,
>
> Authors of paper \#1789

---

> ### Comment · Reviewer_5jDt · 2025-08-05
> **Response**
>
> Thank you for the author's response. This is a great piece of work, and I recommend Borderline accept

---

### Official Review · Reviewer_JU4b · 2025-07-01

**Clarity:** 4
**Significance:** 4
**Originality:** 4
**Rating:** 5
**Confidence:** 5

**Summary:**

This paper introduces VisualQuality-R1, a reasoning-based framework for NR-IQA built upon LMM. The model is trained via reinforcement learning to rank images by visual quality, leveraging the GRPO algorithm used in DeepSeek-R1. A key innovation is the use of a continuous fidelity measure, grounded in the Thurstone model, rather than discretized binary labels as rewards. Experimental results across multiple benchmarks demonstrate that VisualQuality-R1 outperforms existing methods, delivering state-of-the-art performance and producing contextually rich, human-aligned quality explanations.

**Questions:**

Please see the weakness points above.

**Ethical Concerns:**

["NO or VERY MINOR ethics concerns only"]

**Final Justification:**

The response addressed my concerns well, and I would maintain my original score.

The proposed RL2R algorithm is innovative and effective. It would be better if the model could be extended to include more datasets and IQA tasks with different prompts, like Q-Align.

**Limitations:**

yes

**Paper Formatting Concerns:**

n.a.

**Quality:**

4

**Strengths And Weaknesses:**

## Strengths

- The proposed RL2R algorithm is novel and effectively captures the relative nature of image quality assessment. It enables multi-dataset training without requiring perceptual scale alignment.
- The paper conducts extensive experiments, including comparisons with recent methods such as Q-Insight, and demonstrates the superior performance of VisualQuality-R1 across several benchmarks.
- A comprehensive ablation study is presented, clearly analyzing the contributions of different components within the framework.
- Figure 4 is impressive, illustrating the model's progressive improvement over the course of training and providing qualitative insight into its reasoning capabilities.

## Weaknesses

- Given that reasoning is central to the proposed framework, the paper would benefit from more qualitative comparisons with other methods (e.g., Q-Insight), to better showcase the advantages of VisualQuality-R1 in generating interpretable outputs.
- Including typical failure cases would enhance the paper’s transparency and help readers better understand the limitations of the proposed method.
- In Table 4, the performance gap between binary and continuous rewards is relatively small (e.g., 0.772 vs. 0.777 and 0.809 vs. 0.814), which may raise questions about the practical significance of adopting continuous rewards.
- The multi-dataset training setup only includes two datasets. Incorporating a broader range of datasets could strengthen the generalization claims and provide a more thorough evaluation.

## Conclusion

This paper presents a novel approach to NR-IQA through the integration of reasoning with reinforcement learning. The proposed RL2R algorithm  is innovative and effective. The experimental results, including strong baseline comparisons and detailed ablations, support the method’s contributions. Overall, this work represents a significant step forward in the field of IQA, enhances the understanding of IQA and provides a promising direction for future research.

---

> ### Author Rebuttal · Authors · 2025-07-28
>
> We sincerely appreciate this reviewer's insightful feedback and careful review of our manuscript. Please find what below our detailed point-to-point responses to all the comments of this reviewer. We hope our responses can address this reviewer's concerns.
>
> ---
>
> **`[W1] Regarding Qualitative Reasoning Comparison`**
>
> Thanks for the constructive suggestion. We agree that qualitative comparisons can provide valuable insight into the interpretability of model outputs. In the revised version of our manuscript, we will include additional visual examples comparing the reasoning outputs of VisualQuality-R1 with those of Q-Insight and other representative methods to better showcase the strengths of our approach.
>
> Additionally, we have conducted a user study to assess the alignment of VisualQuality-R1’s rationales with human perception of image quality. Please refer to our responses to **`Reviewer NF6n [W3 & Q2]`**  for more details. The results show that 87.5% image-text pairs were judged as aligned. This provides evidence that the reasoning generated by VisualQuality-R1 is largely consistent with human perceptual judgments, despite the absence of explicit supervision at the rationale level.
>
> ---
>
> **`[W2] Regarding Failure Case Demonstration`**
>
> We sincerely thank this reviewer for this insightful suggestion. As also indicated by **`Reviewer 5jDt [Q1 & Q3]`**, our model indeed exhibits some typical failure cases. For example, in the supplementary material, the second image in Figure B produces a score that falls outside the instructed range, and the sixth image in Figure C lacks style-specific reasoning in its explanation. We will include more failure cases in the revised manuscript. These examples will help illustrate scenarios where the model struggles and highlight directions for future improvement.
>
> ---
>
> **`[W3] Regarding Continuous Reward`**
>
> Thanks for the insightful comment. To further investigate the practical significance of continuous rewards over binary rewards, we conducted an additional ablation study under a multi-dataset training setting, which differs from the single-dataset setup used in the main paper. Specifically, we combined four heterogeneous IQA datasets: KADID, TID2013, CLIVE, and SPAQ. The SRCC results are shown in the table below.
>
> \begin{array}{l|cccc|cccc|c}
> \hline
> \text{Method} & \text{BID} & \text{CLIVE} & \text{KonIQ} & \text{SPAQ} &
> \text{Deblurring} & \text{Super-Resolution} & \text{Dehazing} &
> \text{Image Generation} & \text{Avg} \newline
> \hline
> \text{Binary Reward}      & 0.822 & 0.817 & 0.844 & 0.911 & 0.849 & 0.743 & 0.614 & 0.752 & 0.794   \newline
> \text{Continuous Reward} & 0.829  & 0.824  & 0.845  & 0.916 & 0.856  & 0.740  & 0.662 & 0.757  & 0.804 \newline
> \hline
> \end{array}
>
> The results consistently show that continuous reward yields higher PLCC scores on 7 out of 8 tasks, leading to a noticeable +1.0% absolute gain on average. Notably, the improvements are more pronounced on challenging cross-domain settings like Dehazing (+4.8%) and Deblurring (+0.7%), where binary rewards tend to saturate. This suggests that **continuous fidelity rewards provide more informative and fine-grained supervision signals**, helping the policy model better differentiate subtle perceptual quality differences during reinforcement learning.
>
> These findings align with our hypothesis that continuous rewards better reflect the nuanced nature of human perception, which is inherently more graded than binary. As such, even if individual score gaps may seem small, the **consistent directional gain** across diverse tasks and datasets affirms the practical benefit and robustness of the continuous reward design.
>
> ---
>
> **`[W4] Regarding Multi-Dataset Training`**
>
> Thanks for the constructive comment. To address the concern regarding limited dataset diversity in our multi-dataset training setup, we conducted an additional ablation study by combining four heterogeneous IQA datasets (KADID, TID2013, CLIVE, and SPAQ) into a joint training set. This setup introduces greater variation in distortion types, image content, and rating scales, providing a more rigorous evaluation of model generalization. The SRCC results are presented in the table below.
>
> \begin{array}{l|cccc|cccc|c}
> \hline
> \text{Training Dataset} & \text{BID} & \text{CLIVE} & \text{KonIQ} & \text{SPAQ} &
> \text{Deblurring} & \text{Super-Resolution} & \text{Dehazing} &
> \text{Image Generation} & \text{Avg} \newline
> \hline
> \text{KADID}      & 0.790 & 0.750 & 0.830 & 0.875 & 0.838 & 0.756 & 0.598 & 0.775 & 0.777   \newline
> \text{KADID-SPAQ} & 0.811  & 0.811  & 0.855  & 0.913 & 0.845  & 0.752  & 0.588 & 0.754  & 0.791 \newline
> \text{KADID-TID2013-CLIVE-SPAQ} & 0.829  & 0.824  & 0.845  & 0.916 & 0.856  & 0.740  & 0.662 & 0.757  & 0.804 \newline
> \hline
> \end{array}
>
> Compared to training solely on KADID (Avg = 0.777) or KADID-SPAQ (Avg = 0.791), training on all the four datasets further improves the average SRCC to 0.804, demonstrating that VisualQuality-R1 benefits from the increased diversity and volume of training data. Importantly, the performance gains are not isolated to a single dataset or distortion type. For instance, we observe improvements on BID and Dehazing datasets, which are known to pose generalization challenges. This confirms that our **RL2R framework remains effective under broader and more heterogeneous training conditions**. We will include this experiment in the revised manuscript.

---

> ### Author Response · Authors · 2025-08-04
>
> Dear Reviewer JU4b,
>
> Many thanks for your time in reviewing our paper and your constructive comments. We have submitted the point-to-point responses. We appreciate if you could let us know whether your concerns have been addressed, and we are happy to answer any further questions.
>
> Best regards,
>
> Authors of paper \#1789

---

### Official Review · Reviewer_NF6n · 2025-07-02

**Clarity:** 4
**Significance:** 3
**Originality:** 3
**Rating:** 4
**Confidence:** 3

**Summary:**

This paper introduces VisualQuality-R1, a reasoning-induce, no-reference image quality assessment (NR-IQA) model that produces both scalar quality scores (1-5) and natural language justifications. The paper pointed out prior approaches using regression-based approaches which have their limitations and framed IQA as a ranking problem and trained the model using a custom Reinforcement Learning to Rank (RL2R) framework.
The model, built on a VLM (Qwen2.5-VL-7B) generates multiple reasoning-score pairs per image using Group Relative Policy Optimization (GRPO). These are used to compute probabilistic comparisons between image pairs via the Thurstone model, which accounts for both mean and variance of predicted scores. Training feedback comes from a continuous fidelity reward, which measures how well the model’s pairwise rankings align with human preferences (Mean Opinion Scores).
Experiments across eight datasets show that it outperforms existing deep learning and VLM-based IQA methods. It is notable for its ability to generalize across dataset and for producing high-quality justifications for its predictions.

**Questions:**

1. The paper honestly acknowledges that the method is "slow, expensive, and memory-hungry" due to the need to generate multiple responses per image at inference time. Any plan to provide a quantitative comparison of the inference time and computational cost of VisualQuality-R1 versus other leading models?

2. Although the model output rationales, there is no human evaluation or rubric to assess their alignment, usefulness or consistency. What is a plan to move forward with this?

3. Although the paper claims that VisualQuality-R1 can train across datasets without perceptual scale realignment (i.e., without mapping all scores to a common scale), it actually relies on rescaling MOS scores to a common 1–5 range before training. The continuous reward also based on MOS differences assumes access to human scores during training, which limits scalability to unlabeled settings. Can you comment on this?

**Ethical Concerns:**

["NO or VERY MINOR ethics concerns only"]

**Limitations:**

yes

**Quality:**

3

**Strengths And Weaknesses:**

Overall the paper is a comprehensive paper that presents a significant advancement in the field of image quality assessment. The methodology is well motivated and the results are validated. The work is well written and makes a clear contribution.

Strengths:
Novel Approach: Combining GRPO with Thurstone-based quality comparisons and continuous fidelity reward is original within IQA.
Multimodal Reasoning: Demonstrates a step beyond score-only predictions, offering contextual textual explanations.
Comprehensive Evaluation: Benchmarks across eight datasets and various distortion types.
Ablation and Analysis: Includes studies of GRPO response count, reward formulation, and score variance over training.

Opportunities For Improvement:
Computational Cost Analysis: The paper honestly acknowledges that the method is "slow, expensive, and memory-hungry" due to the need to generate multiple responses per image at inference time. While this limitation is stated, a quantitative comparison of the inference time and computational cost of VisualQuality-R1 versus other leading models would provide a clearer picture of the practical trade-offs involved.
Prompt Robustness: The entire methodology relies on a single, fixed text prompt provided in the paper. The authors propose exploring dynamic prompting as future work, but a small-scale study on the current model's sensitivity to minor changes in the prompt wording could have added valuable insight into its robustness
Model's Reasoning Lacks Empirical Substantiation: Although the model output rationales, there is no human evaluation or rubric to assess their alignment, usefulness or consistency.
Overclaim Generalization: Although the paper claims that VisualQuality-R1 can train across datasets without perceptual scale realignment (i.e., without mapping all scores to a common scale), it actually relies on rescaling MOS scores to a common 1–5 range before training. The continuous reward also based on MOS differences assumes access to human scores during training, which limits scalability to unlabeled settings.

---

> ### Author Rebuttal · Authors · 2025-07-27
>
> We sincerely thank this reviewer for the insightful feedback and the thorough evaluation of our manuscript. Please find what below our detailed point-to-point responses to all the comments of this reviewer. We hope our responses can address this reviewer's concerns.
>
> ---
>
> **`[W1 & Q1] Regarding Computational Cost Analysis`**
>
> Thanks for the constructive comment. The concern regarding the computational overhead of the proposed model is fully acknowledged. To address this issue, we have added a quantitative comparison of inference time, GPU memory usage, and FLOPs across several representative NR-IQA models. These results will be included in the revised manuscript.
>
> \begin{array}{l|c|c|c|c}
> \hline
> \text{Method} & \text{Parameters} & \text{Inference Time (BS=1)} & \text{Per-GPU Inference Memory} & \text{TFLOPs} \newline
> \hline
> \text{UNIQUE} & 22.32\text{M} & 0.02\text{s} & 1.62\text{G} & 0.029 \newline
> \text{MUSIQ} & 27.13\text{M} & 0.05\text{s} & 1.69\text{G} & 0.026 \newline
> \text{MANIQA} & 135.75\text{M} & 0.03\text{s} & 2.13\text{G} & 0.217 \newline
> \text{LIQE} & 151.28\text{M} & 0.03\text{s} & 2.15\text{G} & 0.131 \newline
> \text{Q-Align} & 8.20\text{B} & 0.14\text{s} & 17.1\text{G} & 1.98 \newline
> \text{DeQA-Score} & 8.20\text{B} & 0.11\text{s} & 17.1\text{G} & 1.98 \newline
> \text{Q-Insight} & 8.29\text{B} & 2.72\text{s} & 17.6\text{G} & 8.71 \newline
> \text{VisualQuality-R1} & 8.29\text{B} & 2.34\text{s} & 17.6\text{G} & 7.74 \newline
> \hline
> \end{array}
>
> As shown in the table above, VisualQuality-R1 incurs higher inference cost than discriminative models such as MUSIQ and MANIQA, as well as small-scale VLMs like LIQE, primarily due to the large model backbone (Qwen2.5-VL-7B). However, this computational overhead reflects a deliberate design choice aimed at enabling fine-grained, human-aligned reasoning, uncertainty-aware ranking under the Thurstone model, and consistent generalization across diverse distortion scenarios (see Table 2 in the main paper). With the growing popularity and adoption of large-scale VLMs, the increased computational overhead is acceptable.
>
> ---
>
> **`[W2] Regarding Prompt Robustness`**
>
> We appreciate the reviewer’s constructive comment regarding the robustness of fixed prompts. In response, we investigate prompt robustness from two complementary perspectives: 1) introducing new scenario-specific prompt content, and 2) performing controlled perturbations of the original prompt wording. The results will be included in the revised manuscript.
>
> **1) Scenario-Specific Prompt:** As a preliminary step toward addressing this issue, we conducted a small-scale investigation by introducing scenario-specific prompt variants at inference time, without modifying the training configuration. Instead of using a single fixed instruction, we customized the prompt to match the distortion scenario of each test image (e.g., deblurring, super-resolution, dehazing). Specifically, we modified the segment in Table 1 from: *''Rate the overall image visual quality. The rating should be a float between 1 and 5…''* to: *''Rate the overall image visual quality. **[scenario-specific prompt]** The rating should be…''*.
>
> The ***[scenario-specific prompt]*** is dynamically inserted based on the distortion type. Below are examples of the specialized prompts:
>
> - **Realistic scenario:** *The image may contain naturally occurring distortions from digital imaging pipelines, such as noise, blur, compression artifacts, or color inaccuracy. Evaluate the overall perceived quality as a human would perceive it in everyday photography settings.*
> - **Deblurring:** *This image may have undergone a deblurring process. Evaluate the overall visual quality, including sharpness, clarity, and naturalness. Consider whether the deblurring contributes to a visually pleasing result, or if it introduces artifacts, unnatural textures, or over-sharpening effects. Focus on the image as a whole and assess how perceptually high-quality it appears.*
> - **Super-Resolution:** *This image was enhanced by a super-resolution model. Rate its visual quality based on detail, sharpness, coherence, and realism. Watch for artifacts like over-sharpening, unnatural textures, or structural errors.*
> - **Dehazing:** *This is a dehazed image. Assess the overall visual quality after haze removal, focusing on whether the result appears natural, clear, and visually pleasing. Your evaluation should reflect the image’s clarity, realism, and aesthetic coherence as a whole.*
> - **Image Generation:** *This image is generated by an AI model. Evaluate its overall perceptual quality based on both structural coherence and aesthetic appeal. For aesthetics, consider factors such as composition, color harmony, lighting, visual balance, and artistic style.*
>
> Rather than altering the full prompt, we introduced a minimal scenario-specific prefix while keeping the core instruction unchanged. Please refer to our responses to **`Reviewer Cn1F [W5 & Q1]`** for detailed results and analysis. Without modifying training, such prompts bring clear gains for imaging-related distortions, but offer limited improvement for complex processing-related ones, possibly because the model has already captured key cues.
>
> **2) Controlled Perturbations:** We apply prompt perturbations at inference time by randomly introducing minor modifications to the instruction in Table 1, including word substitutions, deletions, and the insertion of special characters. The experimental SRCC results are summarized in the table below. Across all eight datasets, we observe negligible performance differences, suggesting that the model is robust to such minor variations in prompt formulation.
>
> \begin{array}{l|cccc|cccc|c}
> \hline
> \text{Method} & \text{BID} & \text{CLIVE} & \text{KonIQ} & \text{SPAQ} &
> \text{Deblurring} & \text{Super-Resolution} & \text{Dehazing} &
> \text{Image Generation} & \text{Avg} \newline
> \hline
> \text{Original}      & 0.790 & 0.750 & 0.830 & 0.875 & 0.838 & 0.756 & 0.598 & 0.775 & 0.777   \newline
> \text{Perturbed} & 0.791  & 0.747  & 0.829  & 0.875 & 0.838  & 0.751  & 0.599 & 0.776  & 0.776 \newline
> \hline
> \end{array}
>
> ---
>
>
> **`[W3 & Q2] Regarding Empirical Substantiation of Rationales`**
>
> We sincerely appreciate the reviewer’s constructive suggestions. We acknowledge that, to date, there is no well-established metric for accurately evaluating text–image alignment at the low-level quality dimension such as blur, noise, or artifacts, unlike high-level semantic similarity. Hence, we conducted a human study to assess the perceptual alignment of the model’s rationales. We curated a set of 40 test images that were not seen during training, including both natural photographs and outputs from various enhancement algorithms such as deblurring, super-resolution, and AI-generated content. For each image, we collected the model-generated rationale and asked human evaluators to assess whether the reasoning accurately reflects perceptually important quality factors, such as sharpness, contrast, and noise.
>
> Each image-text pair was independently evaluated by 15 domain experts in low-level vision. Following a hard-labeling protocol, we considered a pair to be aligned if at least 10 out of 15 raters deemed the rationale accurate. Each participant took approximately 30 minutes to complete the evaluation.
>
> The results show that 35 out of 40 image-text pairs (87.5%) were judged as aligned. This provides evidence that the reasoning generated by VisualQuality-R1 is largely consistent with human perceptual judgments, despite the absence of explicit supervision at the rationale level. These results will be included in the revised version of the paper.
>
> ---
>
> **`[W4 & Q3] Regarding Overclaim Generalization`**
>
> Thanks for the thoughtful comments. This point touches the core design philosophy of RL2R.
>
> **Rescaling MOS Scores:** We clarify that our method **does not require rescaling** MOS scores to a 1–5 range during training. Rescaling is applied only in Q-Insight’s multi-dataset setup (see footnote, page 7 of our main paper). RL2R’s output is constrained to the 1–5 range solely to align with Q-Insight but this does not imply that input score rescaling is required during training.
>
> As shown in Equation 2, our model relies only on relative comparisons between image quality scores. The continuous reward is based on pairwise preferences, independent of score scale. This relative information can be directly derived from MOS scores in any range (e.g., 0–100 or 1–5), enabling VisualQuality-R1 to train on datasets without perceptual scale realignment.
>
> This design also enables VisualQuality-R1 to utilize datasets that contain only weak or relative supervision, such as BAPPS or DiffIQA, where fine-grained MOS scores are unavailable. Compared to regression-based methods like Q-Insight, which depend heavily on MOS values and require careful normalization, our RL2R formulation is fundamentally more scalable and robust to diverse labeling conditions.
>
> **Generalization to Unlabeled Settings:** Previous work [42] has shown that unlabeled data can be leveraged via gMAD, where image pairs are selected such that a weaker model rates them similarly while a stronger model detects significant perceptual differences. Building on this, we suggest using VisualQuality-R1 as the student and identifying teacher models with superior performance in certain distortion types or content. gMAD can help mine image pairs from unlabeled data where the teacher sees differences that VisualQuality-R1 misses. These pairs can then be used to fine-tune VisualQuality-R1 and enhance its generalization without human labels.
>
> [42] Active fine-tuning from gMAD examples improves blind image quality assessment, TPAMI 2021

---

> ### Author Response · Authors · 2025-08-04
>
> Dear Reviewer NF6n,
>
> Many thanks for your time in reviewing our paper and your constructive comments. We have submitted the point-to-point responses. We appreciate if you could let us know whether your concerns have been addressed, and we are happy to answer any further questions.
>
> Best regards,
>
> Authors of paper \#1789

---

> ### Comment · Area_Chair_7Q1J · 2025-08-07
> **Reviewer NF6n, please check authors' rebuttal**
>
> Reviewer NF6n,
>
> Thanks a lot for your initial review. Please read authors' rebuttal and provide your reply/discussion ASAP before the deadline. Thanks
>
> Your area chair

---

### Official Review · Reviewer_Cn1F · 2025-07-02

**Clarity:** 3
**Significance:** 3
**Originality:** 3
**Rating:** 5
**Confidence:** 5

**Summary:**

The authors propose VisualQuality-R1, a no-reference image quality assessment model that incorporates reasoning through reinforcement learning to rank. Specifically, the method employs GRPO to generate multiple quality estimates per image and leverages the Thurstone model to compute pairwise comparative probabilities. A continuous fidelity reward is introduced to replace traditional binary rewards, aligning better with human perceptual scores. Experimental results demonstrate that the proposed approach outperforms existing SOTA methods across multiple datasets.

**Questions:**

Have the authors conducted further prompt ablation studies, especially evaluating the effect of using different prompts for different distortion types?

**Ethical Concerns:**

["NO or VERY MINOR ethics concerns only"]

**Final Justification:**

My concerns have been adequately addressed in the rebuttal. I would like to elevate my rating to 5 (accept) based on the method's potential for its exceptional scalability and elimination of re-annotation requirements. I recommend open-sourcing the implementation to facilitate validation on larger datasets. This would further verify its scaling capabilities while benefiting the community.

**Limitations:**

Yes

**Quality:**

3

**Strengths And Weaknesses:**

### Strengths
1. The proposed RL2R framework effectively captures the relative nature of human quality perception, as supported by the empirical results in Section 4.
2. The introduction of a continuous fidelity reward, combining predicted probabilities and MOS labels, allows for more informative and stable policy updates in GRPO.
3. Extensive experiments, including evaluations on multiple benchmark datasets and ablation studies, show the method achieves superior performance compared to prior approaches.

### Weaknesses
1. As shown in Table 4, purely relying on relative scoring does not yield significant performance gains. The improvements are mainly observed when enhanced with components like continuous rewards, adaptive variance estimation, and probability averaging, suggesting the base design alone is insufficient.
2. Without joint training on multiple datasets, the proposed method achieves only a marginal improvement over Q-Insight, approximately a 1 percentage point gain, raising questions about its generalizability and robustness in single-dataset scenarios.
3. Given the model's reliance on relative comparisons within a batch, the batch size and sample quality can significantly influence training. It is unclear whether the authors explored the impact of batch construction or optimized sampling strategies.
4. The paper lacks analysis of computational cost, including training/inference time and memory consumption, making it difficult to assess the method's practicality in real-world deployment.
5. The use of a single structured prompt for all distortion types may limit the model’s adaptability to diverse and context-specific quality cues.

---

> ### Author Rebuttal · Authors · 2025-07-27
>
> We are truly grateful for this reviewer's valuable insights and careful evaluation on our manuscript. Please find what below our detailed point-to-point responses to all the comments of this reviewer. We hope our responses can address this reviewer's concerns.
>
> ---
>
> **`[W1] Regarding the Base Design`**
>
> We appreciate the reviewer’s thoughtful concern. However, we respectfully cannot agree that the base design alone is insufficient. To clarify the motivation and demonstrate the effectiveness of our design, which is formulated as pure relative scoring through RL2R, we elaborate on two key aspects below.
>
> 1. As shown in Table 4, VisualQuality-R1, which uses only relative scoring without additional components, achieves SRCC of 0.770 and PLCC of 0.799, comparable to Q-Insight’s 0.766 and 0.803. This demonstrates that the core design is already competitive with existing strong methods, validating its effectiveness and soundness.
>
> 2. Continuous rewards, adaptive variance estimation, and probability averaging are enhancements based on the our relative comparison framework. As shown in Table 4, all variants are built upon this foundation, demonstrating its modularity and extensibility. This flexibility is a result of our framework’s mathematical grounding and statistical expressiveness.
>
> ---
>
> **`[W2] Regarding the Generalizability and Robustness in Single-Dataset Scenarios`**
>
> VisualQuality-R1 and Q-Insight share the same vision-language backbone, Qwen2.5-VL-7B, and they are fully fine-tuned from identical pre-trained weights. The key difference lies in the learning approach: Q-Insight treats IQA as an absolute regression task with binary rewards, while our model adopts a relative formulation using RL2R, applying continuous fidelity-based rewards and uncertainty-aware comparisons based on the Thurstone model. A 1% gain **without architectural changes or pretraining** highlights RL2R’s strength, and consistent improvements across individual datasets confirm its robustness and generalizability in single-dataset settings. We argue that the improvement made by our our method is not marginal.
>
> More importantly, **RL2R is originally designed for multi-dataset training**, where scale misalignment is a major challenge. As shown in Table 4, simply applying Q-Insight’s regression-based objective to multi-dataset training leads to a significant performance drop. In contrast, our RL2R approach not only maintains performance but achieves substantial gains across datasets, further validating its effectiveness in more challenging settings.
>
> ---
>
> **`[W3] Regarding Batch Construction`**
>
> Thanks for the insightful comment. We fully agree that batch construction is important for relative comparison models. We analyzed the impact of per-GPU batch size and sample quality, with SRCC results shown in the table below.
>
> \begin{array}{l|cccc|cccc|c|c}
> \hline
> \text{Method} & \text{BID} & \text{CLIVE} & \text{KonIQ} & \text{SPAQ} &
> \text{Deblurring} & \text{Super-Resolution} & \text{Dehazing} &
> \text{Image Generation} & \text{Avg} & \text{Time} \newline
> \hline
> \text{Q-Insight} & 0.784 & 0.761 & 0.806 & 0.872 & 0.831 & 0.724 & 0.601 & 0.749 & 0.766    & 5\text{h} \newline
> \text{BS=4}      & 0.793 & 0.764 & 0.826 & 0.880 & 0.840 & 0.764 & 0.602 & 0.769 & 0.780  & 8\text{h} \newline
> \text{BS=6}      & 0.793 & 0.768 & 0.830 & 0.881 & 0.841 & 0.755 & 0.607 & 0.764 & 0.780 & 6\text{h} \newline
> \text{BS=8 (Ours)}      & 0.790 & 0.750 & 0.830 & 0.875 & 0.838 & 0.756 & 0.598 & 0.775 & 0.777   & 5\text{h} \newline\hline
> \text{Same Content} &0.744	&0.708	&0.796	&0.857	&0.830	&0.765	&0.536	&0.778	&0.752  & 5\text{h} \newline
> \text{Similar Quality} &\text{fail}  &\text{fail} &\text{fail} &\text{fail} &\text{fail} &\text{fail} &\text{fail} &\text{fail} &\text{fail}   &-  \newline
> \hline
> \end{array}
>
>
> **Impact of Batch Size:** We experimented with per-GPU batch sizes of 4, 6, and 8. We observed that reducing the batch size results in only marginal performance gains (with SRCC improvements within 0.003), but it leads to a substantial increase in training time. For example, the training time increases from 5 hours with BS=8 to 8 hours with BS=4. This inefficiency becomes even more critical under multi-dataset training, where the overall data scale is significantly larger. Therefore, we choose BS=8 to ensure both effectiveness and scalability.
>
> **Impact of Sampling Strategies (Sample Quality):** Our default approach samples images randomly from a single dataset to form each batch. To further investigate the effect of sample quality, we examine two alternative strategies:
>
> 1. Same Content Sampling: We sample batches in which all images share the same content but differ in distortion levels, a strategy applicable only to FR-IQA datasets like KADID-10K. As shown in the table (row: Same Content), this leads to lower performance (SRCC: 0.752), indicating that restricting content diversity weakens generalization. We attribute this to the context-sensitive nature of IQA, where diverse content and distortions offer richer visual cues. In contrast, uniform content may cause overfitting to limited artifacts, which runs counter to our model’s emphasis on semantic reasoning across heterogeneous image pairs.
>
> 2. Similar Quality Sampling: We test a more extreme strategy by sorting images by predicted quality and sampling consecutive ones to reduce intra-batch variance (row: Similar Quality). This leads to training collapse, with the model assigning nearly identical scores to all inputs. The failure highlights the importance of intra-batch diversity for generating effective learning signals in ranking-based frameworks. Diverse content and distortion levels are essential for maintaining visual contrast and enabling robust gradient updates in RL2R. Additionally, exposing vision-language models to a range of quality cases supports stable convergence under continuous reward supervision.
>
> ---
>
> **`[W4] Regarding Computational Cost`**
>
> We sincerely thank this reviewer for the suggestion. As mentioned in Lines 170–171 of the main paper, our training was conducted on 16 NVIDIA A100 GPUs and took approximately 5 hours. To better address the reviewer’s concern, we provide a detailed summary of memory consumption and runtime for both training and inference in the table below.
>
> \begin{array}{lccc}
> \hline
>  \text{Computational Cost} & \text{Training (BS=8, K=6, 10 epochs)} & \text{Inference (BS=32)} & \text{Inference (BS=1)} \newline
> \hline
> \text{Memory} & 68\text{G per GPU} & 23.8\text{G per GPU} & 17.6\text{G per GPU} \newline
> \text{Time} & 5\text{h} & 5.62\text{s (One-Shot)} & 2.34\text{s (One-Shot)} \newline
> \hline
> \end{array}
>
> The table summarizes the memory and time costs for both training and inference, all evaluated on images with a resolution of 512&times;384. Training with batch size 8, K equals 6, and 10 epochs takes about 5 hours and uses 68 GB of GPU memory since the multi-response sampling and pairwise ranking are performed with a 7B model. In inference, the one-shot setting refers to generating a single response per prompt. Under this setting, inference with batch size 32 takes 5.62 seconds and uses 23.8 GB of memory, while batch size 1 takes 2.34 seconds and uses 17.6 GB, which fits within the memory capacity of commonly available GPUs such as the RTX 3090 (24 GB). Please also refer to our responses to **`Reviewer NF6n [W1 & Q1]`** for more details. We will include this information in the revised manuscript.
>
> ---
>
> **`[W5 & Q1] Regarding Single Structured Prompt`**
>
> We sincerely thank this reviewer for the insightful suggestion. We have explicitly listed this as a limitation of our method and highlighted application-aware prompt adaptation as a future direction (please refer to lines 252–256 in our main paper). To further investigate the potential benefit of prompt specialization, we conducted an additional ablation where we kept the training process unchanged and varied the prompt only during inference. Please refer to our responses to **`Reviewer NF6n [W2]`** for the full set of scenario-specific prompts. The updated SRCC results are shown in the table below.
>
> \begin{array}{l|cccc|cccc|c}
> \hline
> \text{Method} & \text{BID} & \text{CLIVE} & \text{KonIQ} & \text{SPAQ} &
> \text{Deblurring} & \text{Super-Resolution} & \text{Dehazing} &
> \text{Image Generation} & \text{Avg} \newline
> \hline
> \text{Fixed Prompt}      & 0.790 & 0.750 & 0.830 & 0.875 & 0.838 & 0.756 & 0.598 & 0.775 & 0.777   \newline
> \text{Diverse Prompt} & 0.795  & 0.767  & 0.840  & 0.887	& 0.837  & 0.742  & 0.588 & 0.778  & 0.779 \newline
> \hline
> \end{array}
>
> The above results reveal several interesting patterns, analyzed as follows:
>
> **Imaging-Related Distortions (BID, CLIVE, KonIQ, SPAQ):** All the four datasets exhibit consistent improvements. This suggests that adding simple textual cues helps the model contextualize low-level distortions such as blur or noise more effectively.
>
> **Processing-Related Distortions (Deblurring, Super-Resolution, Dehazing, Image Generation):** These types of distortions often involve complex trade-offs, such as the trade-offs between fidelity and perceptual quality. In such cases, a short prompt may not provide sufficient guidance, and the model might already capture most of the relevant cues from the image itself, making the additional textual input either unnecessary or potentially confusing.
>
> In this ablation, we only vary the inference-time prompt without retraining the model. Greater gains may be possible by incorporating prompt specialization into training (e.g., via prompt tuning or reward-aware generation), which we leave for future work.

---

> > ### Comment · Reviewer_Cn1F · 2025-08-05
> > **Response**
> >
> > Thanks for the response. It addressed all my questions. I have some additional thoughts for the authors:
> > 1. Could decomposing IQA into finer attributes (e.g., sharpness, lighting) enhance model performance?
> > 2. Do you have recommendations for obtaining accurate labels for these specific attributes?

---

> > > ### Author Response · Authors · 2025-08-05
> > >
> > > Thanks for the thoughtful questions.
> > >
> > > We agree that guiding the model to attend to fine-grained perceptual attributes (e.g., sharpness, lighting, color consistency) can improve performance. As discussed in **`[W5 & Q1]`**, modifying the prompts at inference time led to clear gains in real-world scenarios. Although the effect was less obvious on synthetic distortions, we believe that incorporating attribute-level guidance during training could further enhance generalization. This is part of our planned future work.
> > >
> > > For obtaining accurate attribute labels, we suggest leveraging strong VLMs such as GPT-4o to generate initial labels, which can then be verified or corrected by human annotators. This semi-automatic approach helps reduce annotation cost while maintaining quality, and has shown promise in recent dataset construction practices.

---

> > > > ### Comment · Reviewer_Cn1F · 2025-08-06
> > > > **Comment by Reviewer Cn1F**
> > > >
> > > > The fine-grained direction shows strong practical potential. Thank you for the response. I will increase my score accordingly.

---

> ### Author Response · Authors · 2025-08-04
>
> Dear Reviewer Cn1F,
>
> Many thanks for your time in reviewing our paper and your constructive comments. We have submitted the point-to-point responses. We appreciate if you could let us know whether your concerns have been addressed, and we are happy to answer any further questions.
>
> Best regards,
>
> Authors of paper \#1789

---

### Note · Authors · 2025-08-11

Dear Reviewers and Area Chairs,

We sincerely thank all reviewers for their valuable feedback. We are encouraged that all reviewers have expressed a willingness to maintain or revise toward a positive rating, and we greatly appreciate the constructive spirit throughout the exchange.

We have carefully addressed each question from every reviewer with detailed rebuttals, paying particular attention to concerns regarding computational cost, prompt design during the inference stage, and the alignment of the model’s rationale with user intent. To further clarify our work, we have also conducted and included a substantial number of additional experiments and analyses.

We sincerely hope that these efforts help convey both the insight and the potential impact of our paper, and we are grateful for the reviewers’ valuable feedback that guided these improvements. Thank you again for your time and thoughtful consideration.

Best regards,

Authors of paper #1789

---

### Decision · Program_Chairs · 2025-09-17

**Decision:**

Accept (spotlight)

**Comment:**

This paper introduces VisualQuality-R1, a no-reference image quality assessment (NR-IQA) model to predict quality scores and text justifications, via Reinforcement Learning to Rank (RL2R) framework.

After rebuttal, all reviewers gave positive review scores, and agreeing on advantages like novel topic of enabling reasoning in IQA area,  technically sound methods (RL2R framework, continuous fidelity reward),  etc, comprehensive evaluation, exceptional scalability and elimination of re-annotation requirements,

So I will recommend Accept (spotlight).